# Canine Distemper Outbreak by Natural Infection in a Group of Vaccinated Maned Wolves in Captivity

**DOI:** 10.3390/pathogens10010051

**Published:** 2021-01-08

**Authors:** Vicente Vergara-Wilson, Ezequiel Hidalgo-Hermoso, Carlos R. Sanchez, María J. Abarca, Carlos Navarro, Sebastian Celis-Diez, Pilar Soto-Guerrero, Nataly Diaz-Ayala, Martin Zordan, Federico Cifuentes-Ramos, Javier Cabello-Stom

**Affiliations:** 1Conservation and Research Department, Parque Zoologico Buin Zoo, Panamericana Sur Km 32, Buin 9500000, Chile; vicentevergaraw@hotmail.com (V.V.-W.); nt.diaza@gmail.com (N.D.-A.); martin.zordan@waza.org (M.Z.); 2Departamento de Veterinaria, Parque Zoologico Buin Zoo, Panamericana Sur Km 32, Buin 9500000, Chile; scelis@buinzoo.cl (S.C.-D.); pilar_sotovet@yahoo.com (P.S.-G.); 3Living Collection Unit, Veterinary Medical Center, Oregon Zoo, Portland, OR 97221, USA; carlos.sanchez@oregonzoo.org; 4Faculty of Animal and Veterinary Sciences, University of Chile, Av. Santa Rosa, Santiago 11735, Chile; mjabarcaa@gmail.com (M.J.A.); canavarr@uchile.cl (C.N.); federico.cifuentes@uchile.cl (F.C.-R.); 5World Association of Zoos and Aquariums (WAZA), Carrer de Roger de Llúria, 2, 2-208010 Barcelona, Spain; 6Patagonia Campus, School of Veterinary Medicine, Universidad San Sebastian, Puerto Montt 5480000, Chile; javier.cabello@uss.cl

**Keywords:** canine distemper virus, outbreak, *Chrysocyon brachyurus*, zoo, vaccination

## Abstract

Canine distemper virus (CDV) is one of the most significant infectious disease threats to the health and conservation of free-ranging and captive wild carnivores. CDV vaccination using recombinant canarypox-based vaccines has been recommended for maned wolf (*Chrysocyon brachyurus*) after the failure of modified live vaccines that induced disease in vaccinated animals. Here, we report a CDV outbreak in a captive population of maned wolves that were previously vaccinated. Five juveniles and one adult from a group of seven maned wolves housed in an outdoor exhibit died in April–May 2013 in a zoo in the Metropolitan Region, Chile. Clinical signs ranged from lethargy to digestive and respiratory signs. Diagnosis of CDV was confirmed by histopathology, antibody assays, and viral molecular detection and characterization. The phylogenetic analyses of the nucleotide sequence of the H gene of the CDV genome identified in the two positive samples suggest a close relation with the lineage Europe 1, commonly found in South America and Chile. CDV infections in maned wolves have not been previously characterized. To the authors’ best knowledge, this is the first report of the clinical presentation of CDV in a canine species previously immunized with a recombinant vaccine.

## 1. Introduction

Canine morbillivirus, also known as canine distemper virus (CDV), is the etiological agent of canine distemper (CD). It is a highly contagious disease in dogs and is, along with rabies, the most important infectious disease threat to the health and conservation of free-living and captive wild carnivores [1,2,3]. This virus is present in most countries, and several *Carnivora* families are susceptible, including the *Canidae*, *Procyonidae*, *Mustelidae*, *Hyaenidae*, *Ursidae*, *Viverridae*, *Felidae*, *Ailuridae*, *Phocidae*, and *Otariidae.* It can also affect other mammal orders, like *Cetartiodactyla*, *Primates*, *Rodentia*, and *Pilosa* [4,5,6,7,8,9]. The maned wolf (*Chrysocyon brachyurus*) is the largest member of the *Canidae* family endemic/native to South America. According to the International Union for Conservation of Nature’s (IUCN) Red List of Threatened Species, the maned wolf is considered to be near threatened [10]. The four main threats affecting wild maned wolf populations throughout their distribution range are habitat loss and alteration, human persecution due to livestock losses and cultural beliefs, increasing vehicular traffic in highways resulting in roadkills, and pathogens contracted from domestic animals due to increased contact in disturbed environments [11,12,13,14,15,16].

Canine distemper virus vaccination using recombinant canarypox-based vaccines has been recommended as a preventive protocol for captive maned wolf and other wild carnivore species. When modified live vaccines (MLVs) were used in wild animals, the vaccine virus was shown to cause severe disease in some host species [17,18]. Clinical challenges to determine vaccine efficiency are neither approved nor desired in wild animals; therefore, neutralizing antibody titers are used as a proxy to estimate protection in zoo animals. A limited number of studies have shown that the detection of neutralizing antibodies can be considered protective [19,20]. Although CDV has been historically recognized as a threat to the maned wolf conservation, little is known about the pathological effects on the species [14,16]. To the knowledge of the authors, there are no reports documenting the clinical disease by natural CDV infection in maned wolves. Furthermore, this is the first report of recombinant canarypox-based CDV vaccine failure to protect a wild canid species against the clinical infection produced by this virus and the first CDV outbreak documented in maned wolves (*Chrysocyon brachyurus*).

## 2. Case Presentation

A 9-year-old female maned wolf, part of the collection of a zoological institution in Chile, presented with nonspecific clinical signs in April 2013; these signs included lethargy, sialorrhea, and diarrhea. This female was kept in the same enclosure with a 7-year-old male maned wolf and their five 1-year-old juveniles. Unfortunately, after 6 days of supportive treatment that included intravenous fluid therapy, antibiotics, and vitamin K, the female died. Seven days later, four of the juveniles started showing progressive clinical signs, including lethargy and digestive and respiratory signs. All four juveniles died within 16 days despite supportive treatment (Figure 1). The fifth juvenile (a female) died acutely 9 days after her mother without any previous clinical signs. The 7-year-old male remained clinically healthy during the outbreak.

All five juveniles had been vaccinated for the first time with a single dosage 9 months prior to the outbreak (July 2012), and both adults were vaccinated with their first single dosage 5 months prior to the outbreak (November 2012). Before this event, both adults had been vaccinated only against rabies. The vaccine used in all cases were single doses of 1 mL of Recombitek C6 vaccine (Boehringer Ingelheim GmbH, imported and distributed by Boehringer Ingelheim Chile Ltda., Isidora Goyenechea 3000, Las Condes, Santiago, 832000, Chile) administered subcutaneously (SC) in the interscapular space. The vaccines were kept at 4 °C prior to their use. All the animals were in apparent good health status and had shown adequate body weight and normal appetite prior to the outbreak. It is important to mention that the vaccine used is the only CDV recombinant vaccine available in South America.

## 3. Results

### 3.1. Necropsy

In all cases, the most relevant macroscopic findings were moderate pulmonary edema and mild multifocal intestinal erosions. Due to financial constraints, tissue samples from only two juveniles/individuals were collected for histologic evaluation. A full set of tissues from two juveniles was submitted in 10% buffered formalin for histologic evaluation.

### 3.2. Histopathology

The histological evaluation in both cases showed severe diffuse (bronchointerstitial) lymphocytic and histiocytic pneumonia with the presence of intracytoplasmic eosinophilic inclusion bodies. Similar inclusion bodies were found in the kidney, digestive epithelium, trachea, liver, lymph nodes, and urinary bladder. Other findings include diffuse congestion of the liver, atrophy, and moderate to severe diffuse lymphoid depopulation on the lymph nodes and spleen. In one of the two animals, moderate focal ulcerative gastritis was found. The remaining samples did not show pathological changes.

### 3.3. Antibody Detection Assays

The six individuals from the outbreak that were tested with the immunofluorescence assay (IFA) yielded positive results for the presence of IgM and IgG, and the two animals from the control population tested negative for IgM and IgG.

Samples from the six animals that tested positive by IFA were then tested with a serum neutralization (SN) assay. Four out of the six animals had titers ≥1:16, with the remaining two animals showing titers below 1:16, our predetermined threshold [2]; the second population was not tested by SN.

### 3.4. Virus Detection and Characterization

Positive results were also obtained via qPCR in the six samples from the outbreak population, and the two animals from the control population tested negative.

Four of the positive samples were tested by RT-PCR to amplify the H gene. The H gene was successfully amplified and sequenced from two of these samples (4517 and 7490), and the sequences were submitted to GenBank (“Chile/lobo crin/1” (4517) GenBank accession number MW358494 and “Chile/lobo crin/2” (7490) GenBank accession number MW358495).

## 4. Discussion

Infectious diseases are an important threat to the health and conservation of free-ranging wildlife and those under human care; in the latter scenario, infectious diseases can even put at risk the success and viability of breeding programs. CDV has been identified as one of the most significant diseases of wild carnivores in zoos [21], making the implementation of vaccination protocols strongly recommended [22]. This manuscript describes the first report of this naturally acquired disease in maned wolves, which culminated in the death of six vaccinated animals. As such, this represents a new challenge in CDV vaccination protocols for wild canid species.

The pathological, immunohistochemical, and molecular findings observed in maned wolves are consistent with those previously described in several species infected with CDV [23]. The phylogenetic analyses of the nucleotide sequence of the H gene of the CDV genome identified in the two positive samples suggest a close relation with the lineage Europe 1, commonly found in South America [24] and Chile (unpublished).

It is necessary to discuss the fact that the surviving male adult had results that indicated it was infected but never manifested any clinical signs, suggesting a subclinical infection. Unfortunately, it was not possible to obtain a more specific SN titer or to have a Ct value for the qPCR due to the nature of the assays.

The potential route by which the virus might have been introduced to the group of captive maned wolves remains unclear. Pets are not allowed inside the zoo, and the only species of small free-ranging carnivores present in the area are feral cats. Since this outbreak, all susceptible species have been systematically vaccinated with a three-initial-dosage protocol in offspring, plus a single yearly booster thereafter. The zoo collection has been free of CDV cases since this present report.

CDV has been described as one of the most important infectious diseases affecting maned wolves in captivity; however, this conclusion has been solely based in serological studies or clinical signs [16,25,26,27,28], without having a complete correlation of clinical, pathological, and molecular evidence in every report. Without this triad, it is challenging to rule in the clinical causality of a pathogen over a host [29,30].

The current recommendation by the Infectious Diseases Manual [31] of using a recombinant canarypox-vectored vaccine is based on the evidence that canine MLV can result in the reversion of vaccine strains to virulent virus and vaccine-induced CDV in non-domestic carnivores. Inactivated CDV whole-virus vaccines do not produce sufficient immunity to prevent infection after virus challenge [12,21]. The six maned wolves that ultimately died from this outbreak were vaccinated with one dosage of Recombitek within the previous year. Although this is the first report of recombinant CDV vaccine failure to protect a wild canid species against fatal CDV infection, it has previously been documented in snow leopard (*Panthera uncia*) despite prior one-dose vaccination with the monovalent canine rCDV vaccine 3 months prior [32].

A recently published paper [33] supports our findings by showing that a single-dosage protocol of Recombitek CDV vaccine did not elicit a measurable humoral immune response. As further shown in the current report, low SN titers from a single dose of Recombitek CDV vaccine are nonprotective, as evidenced by the fatal outcome in six of the seven vaccinated animals.

To the best of our knowledge, this is the first report of the clinical presentation of CDV in a canine species previously immunized with a canarypox-based CDV vaccine. The effectiveness of protection against CDV with a single dose of this type of vaccine (canarypox vectored) has recently been challenged by different studies, where the humoral and cellular immune responses were measured in red fox (*Vulpes vulpes*) and giant panda (*Ailuropoda melanoleuca*), respectively, with results that suggested limited protective benefits in these species [2,34]. Facing this new body of evidence, it is clear that there is a need for new studies of cellular and humoral immune responses to vaccination in this and other species of wild carnivores.

The present study confirms the lethal nature of CDV in maned wolf and reinforces the necessity for preventive medicine protocols that include the vaccination of this species in captivity. The development of new studies on wild populations is imperative to understand the role of this pathogen in the maned wolf conservation and population dynamics. Finally, the protection failure of the vaccine, when administered as one dose, against CDV shown in this report makes necessary new studies that could present new protocols that improve the effectiveness of its use in this and other species, perhaps contemplating additional booster dosages.

## 5. Materials and Methods

### 5.1. Control Population

In addition to the family group where the outbreak took place, there were two additional 2-year-old maned wolves housed at a different area, approximately 1 km away. These animals were clinically healthy during the outbreak but were tested as a control population.

### 5.2. Necropsy

A complete necropsy was performed soon after the death of five of the six animals (Figure 1).

### 5.3. Immunofluorescence Assay (IFA)

Serum from the six animals (Figure 1) was tested for CDV IgG and IgM. MegaScreen^®^ was used per the supplier’s instructions. For the evaluation, a fluorescence microscope with a filter system for FITC (excitation wavelength, 465–495; barrier filter, 515–555) and 400× magnification was used. A fluorescent intensity ≥1:40 was determined to be positive.

### 5.4. Serum Neutralization Assay

Serum from the six animals (Figure 1), which had been stored for 10 days at −20 °C, was tested for CDV antibodies. The serum was diluted and mixed with a fixed dose of a native viral strain of CDV, previously isolated at the Virology Laboratory of the Agricultural and Livestock Service (SAG) of Chile. The cellular substrate for the test was based on MDCK cells, American Type Collection, from the renal cortex. After 72 h of incubation in an atmosphere of 5% CO_2_, 37 °C, and 95% humidity, the samples were titered by reading the cytopathic effects [35,36]. Titers ≥ 1:16 of the neutralizing antibodies were considered positive [2].

### 5.5. Nucleic Acid Extraction and PCR Assays (qPCR and RT-PCR)

qPCR: 2 μL of viral RNA was extracted from serum of the six animals (Figure 1) using the TRIzol reagent according to the manufacturer’s instructions (Invitrogen, Carlsbad, CA, USA). Reverse transcription was performed using the Lig™ RNAse Inhibitor Solution and Bio™ Transcriptase Solution, part of the VetqPCR-realtime™ CDV Real Time Kit, according to the manufacturer’s instructions (Bioingentech Ltd., Concepcion Chile). A single well is used for qualitative analysis, and a Ct value ≤ 40 is positive, according to the manufacturer’s instructions. The reverse transcription uses 2 cycles, one for initial denaturation (at 65 °C for 10 min) and another to stop the process (at 4 °C for 5 min). To obtain cDNA from the samples, 3 cycles were used: annealing (at 25 °C for 10 min), extension (at 35 °C for 60 min), and denaturation (at 70 °C for 10 min). Lastly, for real-time cycling, a single cycle for initial denaturation was performed (at 94 °C for 2 min), and then 45 cycles of denaturation and annealing were run (at 95 °C for 15 s and at 60 °C for 60 s, respectively).

RT-PCR: Blood samples from four animals were tested with RT-PCR. Viral RNA extraction was performed using the TRIzol reagent according to the manufacturer’s instructions (Invitrogen, Carlsbad, CA, USA). Reverse transcription PCR was performed using an Apollo thermocycler (CLP, Progen Scientific Ltd., London UK) with 96 wells of 0.2 mL each and the SuperScript III One-Step RT-PCR System with Platinum Taq DNA Polymerase Kit plus oligonucleotide primers CDV1 (forward) and CDV2 (reverse) targeting the H gene, according to the manufacturer’s instructions (Invitrogen, Carlsbad, CA, USA) [37].

RT-PCR-positive samples were sent to a local sequencing center (Genytec Ltda., Santiago, Chile) for sequencing in triplicate. The resulting nucleotide sequences were aligned using the ClustalW program, and then compared with all sequences deposited in GenBank using the BLAST program to identify the origin of these amplified DNA fragments.

Pairwise and multiple sequence alignments at the nucleotide and amino acid levels and sequence similarities were calculated using the MEGA v6 software (see [35] for more details about RT-PCR).

## Figures and Tables

**Figure 1 pathogens-10-00051-f001:**
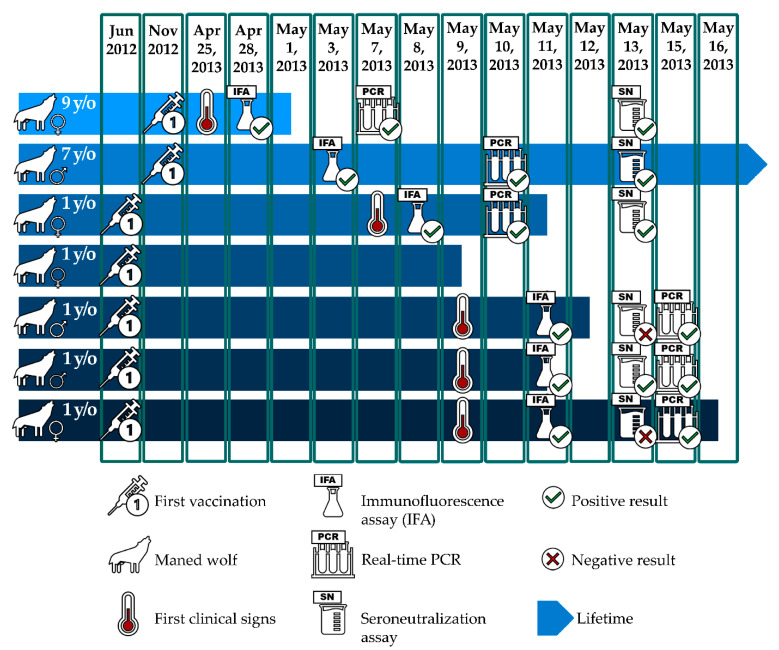
Timeline. Conceptual scheme that depicts the chronological events since first vaccination. It describes the first presentation of clinical signs (if any), the moment of death, and the time when the diagnostic tests were performed.

## Data Availability

Data sharing not applicable. No new data were created or analyzed in this study.

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
