# Peer review of "Canine Distemper Outbreak by Natural Infection in a Group of Vaccinated Maned Wolves in Captivity"

_pathogens, 2021, doi:10.3390/pathogens10010051_

Round 1
Reviewer 1 Report
This manuscript is a valuable contribution to science. The information contained in the manuscript will likely impact how maned wolves, and perhaps other wolf species, are cared for resulting in modified international vaccination guidelines for this species.
There are a few typos that are distracting. Several of the citations do not appear correct and Megacor, 2006 is not listed within the references. Attention to detail throughout the manuscript could improve the clarity.
The authors do not give the vaccination history for the 7 and 9 yr old adult wolves that were exposed to the CDV. Did these wolves only receive a single Recombitek6 vaccine? Providing the full vaccination history may help the readers understand the different clinical response to exposure.
The authors indicate the pups that died of CDV were only vaccinated once with the Recombitek6 vaccination? The package insert recommends adult domestic dogs, or pups over 12 weeks of age receive primary vaccination then a booster/revaccination with a second 1 mL dose 2 to 3 weeks later. Dogs younger than 12 weeks of age should be revaccinated with a single 1 mL dose every 2 to 3 weeks, the last dose given at or over 12 weeks of age. Revaccinate annually with a single 1 mL dose.
Vaccinations also have specific storage guidelines. Were these observed for the vaccine used as this could impact the efficacy of the vaccine?
Figure 1 is a nice image. It may be possible to combine table 1 and figure 1 for an even stronger figure/manuscripts. Generally speaking a figure (or table) should be able to stand alone so that the reader has all necessary information to evaluate the findings. There are not many details within either the table or figure for this manuscript. A figure legend with appropriate information should be added. Information to consider within the legend should include, detailing the virus evaluated (e.g. CDV), relevant methodology, all previous vaccination of adults, species evaluated (e.g. maned wolves), the vaccine used (e.g. Recombitek6), tissues sampled (i.e. serum, necropsy tissues). It would be interesting to include the information on how long the animals showed clinical signs before death.
Line 85 – Suspect there is a typo, perhaps should read “not in sequential order”? Currently reads, “not in sequential other”.
Line 138 – “Unspecific” should be “non-specific”
Line 143 – 144 – Suggest restructuring this sentence, it is long and difficult to follow. Remove the term “a lapse”?
Line 178 - “After 72 hrs. of incubation on an 178 atmosphere of 5% CO2, 37°C and 95% humidity, the samples were titled”. Should this be “titered”?
Author Response
Dear Reviewer 1
Thanks by your comments. We appreciate your recommendations. We attempted to include most comments, and we provide a justification for each case were we decided not to include the recommendation. Below you can find our detailed response. We hope the revised version (Please see the attachment) may be now suitable for publication.
There are a few typos that are distracting. Several of the citations do not appear correct and Megacor, 2006 is not listed within the references. Attention to detail throughout the manuscript could improve the clarity.
Accepted. The citations were reviewed and arranged.
The authors do not give the vaccination history for the 7 and 9 yr old adult wolves that were exposed to the CDV. Did these wolves only receive a single Recombitek6 vaccine? Providing the full vaccination history may help the readers understand the different clinical response to exposure.
Authors Correction, the adult animals had never received CDV vaccinations before. The couple of adult maned wolves arrived to the zoo in 2009 with non previous vaccination. It was 2011 when the recombinant vaccine (Rocombitek C6) was available for the first time in the country.
The authors indicate the pups that died of CDV were only vaccinated once with the Recombitek6 vaccination? The package insert recommends adult domestic dogs, or pups over 12 weeks of age receive primary vaccination then a booster/revaccination with a second 1 mL dose 2 to 3 weeks later. Dogs younger than 12 weeks of age should be revaccinated with a single 1 mL dose every 2 to 3 weeks, the last dose given at or over 12 weeks of age. Revaccinate annually with a single 1 mL dose.
According to the manufacturer's information (copied below the manufacturer's website): https://www.sudamerica.boehringer-ingelheim.com/salud-animal/productos/argentina/animales-de-compania/recombitek-c6
“In previously unvaccinated dogs older than 12 weeks of age, the vaccination plan is at the discretion of the veterinarian. It is recommended to revaccinate annually with a dose of 1 ml”
The management of adults was done following these recommendations. It is important to emphasize that the first animal that suffered clinical CDV and died was the female; she was vaccinated according to the manufacturer's recommendations.
Vaccinations also have specific storage guidelines. Were these observed for the vaccine used as this could impact the efficacy of the vaccine?
The following manufacturer's specifications were followed for correct storage: Between 2 and 7 ° C. Protect from light. Do not freeze. Taked from the manufacturer's website.
Figure 1 is a nice image. It may be possible to combine table 1 and figure 1 for an even stronger figure/manuscripts. Generally speaking a figure (or table) should be able to stand alone so that the reader has all necessary information to evaluate the findings. There are not many details within either the table or figure for this manuscript. A figure legend with appropriate information should be added. Information to consider within the legend should include, detailing the virus evaluated (e.g. CDV), relevant methodology, all previous vaccination of adults, species evaluated (e.g. maned wolves), the vaccine used (e.g. Recombitek6), tissues sampled (i.e. serum, necropsy tissues). It would be interesting to include the information on how long the animals showed clinical signs before death.
Accepted. A new image was included in the page 3 with information combined.
Line 85 – Suspect there is a typo, perhaps should read “not in sequential order”? Currently reads, “not in sequential other”.
This table and text was deleted of Ms.
Line 63 – “Unspecific” should be “non-specific” Accepted.
Line 143 – 144 – Suggest restructuring this sentence, it is long and difficult to follow. Remove the term “a lapse”? Accepted. Changes in the lines 68 to 70: “Clinical signs in the remaining four pups were characterized by lethargy, digestive and respiratory signs. All four pups died within 16 days in spite of treatment (Figure 1)”
Line 180 - “After 72 hrs. of incubation on an 178 atmosphere of 5% CO2, 37°C and 95% humidity, the samples were titled”. Should this be “titered”? Accepted.
Reviewer 2 Report
Vergara-Wilson et al describe a CDV outbreak in a group of vaccinated captive maned wolves in Chile. The observation is worthwhile reporting, but the manuscript requires extensive editing before it can be considered for publication.
- Line 19 and line 36: remove “Canine morbillivirus, also known as”. The virus name is canine distemper virus.
- Line 20: change ‘diseases threat’ to ‘disease threats’
- Line 21: change to “CDV vaccination using recombinant canarypox-based vaccines has…”.
- Line 24: delete: ‘with a recombinant vaccine’.
- Line 37: delete ‘in which elevated mortality and morbidity’.
- Line 50: change ‘dog commercial’ to ‘canarypox-based’.
- Line 52: change ‘some of them induced disease’ to ‘the vaccine virus was shown to cause severe disease in some host species’.
- Line 54: remove ‘by ethical reasons’. The authors should note that validation of the vaccination protocol by measuring virus neutralizing antibodies in serum would have been possible, and for morbilliviruses detection of neutralizing antibodies is usually considered a good correlate of protection. I do understand that this is not easy and not cheap, but in my opinion it would balance the suggestion that it is not possible to evaluate the efficacy of a vaccine in a new species (see also discussion, line 125).
- Line 61: start the result section with the description of the outbreak (now lines 136-165 and figure 1). Without this description it is impossible the understand and interpret the other results. Moreover, an outbreak description is not a material nor a method.
- Line 137: delete ‘was’.
- Line 140: it is confusing to me that the young animals are referred to as pups, but elsewhere as juveniles. Consider rephrasing.
- Line 144: change ‘range’ to ‘ranged’.
- Line 150: the authors describe that bloodwork was performed, but do not report the results. I would encourage to include the data, if not the remark about bloodwork should be removed. Were the animals lymphopenic? In addition, the authors should describe if any serum samples had been stored from previous years to retrospectively check if the adult animals had VN antibodies.
- Line 155: the authors refer to an annual booster: they should describe when the primary vaccination was given and how many boosters the adult animals had received during their lifetime.
- Line 159-160: it is also important to note that the Merial vaccine is (as far as I know) no longer available.
- Line 74: abbreviation SN should be introduced at first use.
- Line 81/82: GenBank accession numbers should be provided for the sequences obtained.
- Line 91: change ‘breading’ to ‘breeding’.
- Line 102: change ‘sings’ to ‘signs’.
- Line 107: add ‘vaccine’ aftyer ‘canarypox-vectored’, remove ‘commercially available’, change ‘cause’ to ‘result in’.
- Line 111-113: of two adults with a history of multiple vaccinations, one animal died and one was protected. The juvenile animals only received one vaccination at the age of 3 months. This difference should be pointed out: I am not sure what the formal recommendation for minimum age at first vaccination is, or how many does are recommended. I find it less surprising that these young animals proved to be unprotected than that the mother got infected and died.
- Line 115: change ‘the clinical’ to ‘fatal CDV’.
- Line 121: change ‘recombinant vaccine’ to ‘canarypox-based CDV vaccine’.
- Lines 125-127: see also my comment 15 – is the vaccine still available?
- Discussion: the authors should briefly discuss potential routes by which the virus may have been introduced to the group of captive maned wolves. I suspect either by pets brought on by visitors or by small free-ranging carnivores. Do the authors believe that prevention of exposure is a feasible additional measure to be recommended?
- Lines 171-172: the authors should explain how positivity was defined in this assay, it is not clear to me what they mean by ‘a fluorescent intensity >1:40.
- Line 174: change ‘challenged to a fix concentration’ to ‘mixed with a fixed dose’. The authors should explain how long the serum/virus mixture was incubated (at 37 degrees?) before adding to the cells.
- Line 178: explain ‘the samples were titled’. The cytopathic effects were read?
- Line 188: explain which primers were used, either by citing a reference or by describing the sequence of the primers. Which viral gene was targeted by the PCR?
Author Response
Dear Reviewer 2
Thanks by your comments. We appreciate your recommendations. We attempted to include most comments, and we provide a justification for each case were we decided not to include the recommendation. Below you can find our detailed response. We hope the revised version (Please see the attachment) may be now suitable for publication.
- Line 19 and line 36: remove “Canine morbillivirus, also known as”. The virus name is canine distemper virus. Accepted.
- Line 20: change ‘diseases threat’ to ‘disease threats’. Accepted.
- Line 21: change to “CDV vaccination using recombinant canarypox-based vaccines has…” Accepted.
- Line 24: delete: ‘with a recombinant vaccine’. Accepted.
- Line 37: delete ‘in which elevated mortality and morbidity’. Accepted.
- Line 50: change ‘dog commercial’ to ‘canarypox-based’. Accepted.
- Line 52: change ‘some of them induced disease’ to ‘the vaccine virus was shown to cause severe disease in some host species’. Accepted.
- Line 54: remove ‘by ethical reasons’. The authors should note that validation of the vaccination protocol by measuring virus neutralizing antibodies in serum would have been possible, and for morbilliviruses detection of neutralizing antibodies is usually considered a good correlate of protection. I do understand that this is not easy and not cheap, but in my opinion it would balance the suggestion that it is not possible to evaluate the efficacy of a vaccine in a new species (see also discussion, line 125).
We thanks the reviewer for this comment. The phrase: “taking in consideration that as wild species it’s not ethically viable to perform challenge studies” was removed from the manuscript. Instead the following was added to the manuscript to address the comment from the reviewer: “Clinical challenges to determine vaccine efficacy is not approved nor desired in wild animals; as such measurements of neutralizing antibodies are used as proxy to estimate protection in zoo animals. A limited number of studies have shown that the detection of neutralizing antibodies can be considered protective (Bronson et al. 2007, Connolly et al 2013),
- Line 61: start the result section with the description of the outbreak (now lines 136-165 and figure 1). Without this description it is impossible the understand and interpret the other results. Moreover, an outbreak description is not a material nor a method.
Accepted.
- Line 137: delete ‘was’. Accepted.
- Line 140: it is confusing to me that the young animals are referred to as pups, but elsewhere as juveniles. Consider rephrasing.
We thanks the reviewer for noticing this inconsistency; we have now used only pups in the manuscript. Based on the age of the younger animals they are considered pups using nomenclature generally accepted for domestic canids where pups are animals of less than 15 months of age.
- Line 144: change ‘range’ to ‘ranged’. Accepted.
- Line 150: the authors describe that bloodwork was performed, but do not report the results. I would encourage to include the data, if not the remark about bloodwork should be removed. Were the animals lymphopenic? In addition, the authors should describe if any serum samples had been stored from previous years to retrospectively check if the adult animals had VN antibodies
The remark about the bloodwork was removed. Unfortunately there are no stored serum samples from previous years for the adult wolves.
- Line 155: the authors refer to an annual booster: they should describe when the primary vaccination was given and how many boosters the adult animals had received during their lifetime.
Authors Correction, the adult animals never had received CDV vaccinations before. The couple of adult maned wolves arrived to the zoo in 2009 with non previous vaccination. It was not only 2011 when the recombinant vaccine (Rocombitek C6) was made available for the first time in Chile
- Line 159-160: it is also important to note that the Merial vaccine is (as far as I know) no longer available.
The vaccine is still available but it is now manufactured by Boehringer Ingelheim GmbH., and imported and distributed by Boehringer Ingelheim Chile Ltda. Isidora Goyenechea 3000, Las Condes, Santiago, 832000, Chile.
- Line 74: abbreviation SN should be introduced at first use. Accepted.
- Line 81/82: GenBank accession numbers should be provided for the sequences obtained. The GenBank accession numbers for our nucleotide sequences are:
BankIt2407908MW358494
BankIt2407913 MW358495
- Line 91: change ‘breading’ to ‘breeding’. Accepted.
- Line 102: change ‘sings’ to ‘signs’.Accepted.
- Line 107: add ‘vaccine’ aftyer ‘canarypox-vectored’, remove ‘commercially available’, change ‘cause’ to ‘result in’. Accepted.
- Line 111-113: of two adults with a history of multiple vaccinations, one animal died and one was protected. The juvenile animals only received one vaccination at the age of 3 months. This difference should be pointed out: I am not sure what the formal recommendation for minimum age at first vaccination is, or how many does are recommended. I find it less surprising that these young animals proved to be unprotected than that the mother got infected and died.
We have in the text (Line 203-204) a Disclaimer: The couple of adult maned wolves arrive to the zoo in 2009. It was 2011 when the recombinant vaccine (Rocombitek C6) was available for the first time in the country. While the pups had only received one vaccine at 3 months of age. We hope this clarification makes it clearer for the reader.
- Line 115: change ‘the clinical’ to ‘fatal CDV’. Accepted.
- Line 121: change ‘recombinant vaccine’ to ‘canarypox-based CDV vaccine’. Accepted.
- Lines 125-127: see also my comment 15 – is the vaccine still available?
Yes, the vaccine is still available in Chile and several other countries in South America (Argentina, Colombia, and Bolivia)
- Discussion: the authors should briefly discuss potential routes by which the virus may have been introduced to the group of captive maned wolves. I suspect either by pets brought on by visitors or by small free-ranging carnivores. Do the authors believe that prevention of exposure is a feasible additional measure to be recommended?
Accepted. Included in Line 132 to 136 “The potential route by which the virus may have been introduced to the group of captive maned wolves remains unclear, the more probable with access to the area of Maned wolf facility are feral cats. Since this outbreak, all susceptible species had been systematically vaccinated with a 3 initial dosage protocol in offspring, plus a single yearly booster thereafter. The zoo collection has been free of CDV cases since this present report”
- Lines 171-172: the authors should explain how positivity was defined in this assay, it is not clear to me what they mean by ‘a fluorescent intensity >1:40.
Accepted. Included Line 171 to 174: “Serum from 6 animals was tested for CDV IgG and IgM MegaScreen® was used per supplier's instructions. For the evaluation, a fluorescence microscope with a filter system for FITC (excitation wavelength 465–495, barrier filter 515–555) and 400× magnification was used. A fluorescent intensity ≥1:40 was determined to be positive”
- Line 174: change ‘challenged to a fix concentration’ to ‘mixed with a fixed dose’. The authors should explain how long the serum/virus mixture was incubated (at 37 degrees?) before adding to the cells and Line 178: explain ‘the samples were titled’. The cytopathic effects were read?
Accepted. Included Line 178-180: “The cellular substrate for the test was base in MDCK cells, American Type Collection, from renal cortex. After 72 hrs. of incubation on an atmosphere of 5% CO2, 37°C and 95% humidity, the samples were titered by reading the cytopathic effects (Hidalgo-Hermoso, Cabello, et al. 2020, Millán, et al. 2016).”
- Line 188: explain which primers were used, either by citing a reference or by describing the sequence of the primers. Which viral gene was targeted by the PCR?
Accepted. Included Line 188 to 190.
Round 2
Reviewer 2 Report
The authors have adequately addressed the comments of the reviewers.